# Odontogenic Sinusitis from Classical Complications and Its Treatment: Our Experience

**DOI:** 10.3390/antibiotics12020390

**Published:** 2023-02-15

**Authors:** Lorenzo Sabatino, Michelangelo Pierri, Francesco Iafrati, Simone Di Giovanni, Antonio Moffa, Luigi De Benedetto, Pier Carmine Passarelli, Manuele Casale

**Affiliations:** 1Unit of Integrated Therapies in Otolaryngology, Fondazione Policlinico Universitario Campus Bio-Medico, Via Alvaro del Portillo, 00128 Rome, Italy; 2Unit of Integrated Therapies in Otolaryngology, Department of Medicine and Surgery, Università Campus Bio-Medico di Roma, Via Alvaro del Portillo, 00128 Rome, Italy; 3Department of Head and Neck, Division of Oral Surgery and Implantology, Institute of Clinical Dentistry, Università Cattolica del Sacro Cuore, Fondazione Policlinico Universitario Gemelli, 00128 Rome, Italy

**Keywords:** odontogenic sinusitis, classic complications

## Abstract

Odontogenic sinusitis (ODS) refers to the maxillary sinus infection, which is secondary to either adjacent infectious dental pathologies or procedures. The aim of this retrospective study is to report the experiences of the department of integrated therapies in otolaryngology (Campus Bio-Medico Foundation, Rome, Italy) in classifying and treating patients that are affected by odontogenic sinusitis derived from “classic complications”. A total of 68 patients responding to the criteria and to the definition as a classical odontogenic complication were included. The surgical therapy consisted of a combined oral and nasal simultaneous approach for 28 patients (43%), a combined non-simultaneous approach for 4 patients (6%), a nasal only approach for 14 patients (21%), and an oral only approach for 20 patients (30%). All the patients presented a complete resolution of the symptoms. The choice of performing a nasal, oral, or combined approach is based on the presence of anatomical elements that facilitate sinusitis and reinfection occurrence, such as deviated nasal septum, concha bullosa, or obstructed osteo-meatal complex. The correct use of validated classification, the pre-operative CT scan, a multidisciplinary approach, and an appropriate presurgical examination are the necessary elements to obtain a good success rate.

## 1. Introduction

Sinusitis can be defined as the inflammation of the nasal sinus mucosa, and clinically, it is identified as the presence of a nasal obstruction, anterior or posterior rhinorrhea ± pain or facial pressure, and hyposmia or anosmia [1]. There are different methods to classify this disease using etiology, timing [2], symptomatology duration, and triggers [3].

Chronic rhinosinusitis (CRS) is defined as the presence of pathology and symptoms for more than 3 months despite therapy. CRS prevalence in Europe is reported to be 10.9% of the general population in the age group between 15 and 75 years of age [4], determining a socio-economic cost both in terms of treatment and missing work days [5]. What is pivotal in successfully treating CRS is understanding its pathophysiology. One of the least known types of CRS is the odontogenic sinusitis (ODS) [6]. ODS refers to the maxillary sinus infection, which is secondary to either adjacent infectious dental pathology, or procedures [7]. It is estimated that ODS accounts for 25–40% of all chronic maxillary sinusitis, most commonly unilateral [8].

ODS is generated by the dynamic interaction of the maxillary sinus and the upper teeth, and this relationship evolves over time and has a great interindividual variability [7]. Infections of a dental origin, when a favorable anatomy is present, can violate the Schneiderian membrane, the mucosal membrane covering the maxillary cavity, and trigger the mucosal inflammatory response [7]. The Schneiderian membrane violation can be the result of several conditions, the most common seems to be iatrogenic injuries (dental implants with dimensions and insertion axis not adapted to the individual clinical features, incorrectly performed sinus lift procedures, foreign bodies, etc.), and others are ascending dental infections, sinus involving odontogenic cysts, and maxillary bone traumas [7]. The most common tooth associated with sinus pathology is the first molar, followed by the second and third [7].

Felisati et al. in 2013 classified the sino-nasal complication into three groups: the first group is composed of odontogenic sinusitis started before an implantological treatment, the second group of odontogenic sinusitis started after an implantological treatment, and the third group is composed of classical odontogenic complications of dental disease and treatment. The classical odontogenic sinusitis is considered an infectious condition of the paranasal sinuses following dental diseases or treatments, including pulpal necrosis in deep caries, periodontitis, odontogenic cysts, endodontic procedures and tooth extractions, with or without the presence of oroantral communication (OAC), and the sinus penetration of endodontic material, or dental fragments. This group can be distinguished into two conditions: bacterial or fungal sinusitis with or without OAC [9].

The teeth most commonly associated with OS are the molars, while extraction has been reported as the principal etiological factor [10].

Odontogenic sinusitis often has a non-specific presentation, but when not addressed properly, it can lead to serious complications, such as extensions to the cranial structures or orbital cavity [7]. The treatment differs from what is employed in the maxillary sinusitis with rhinologic etiology and often requires a multidisciplinary approach involving the ear, nose and throat (ENT) specialist, Maxillofacial surgeon, who is both a dentist and radiologist [2,11].

The first line of therapy is medical therapy, consisting of at least two cycles of oral antibiotics (mostly penicillin, but also cephalosporins and fluoroquinolones are reported), and nasal irrigation. [1,12,13]. Medical therapy is used to alleviate symptoms during the waiting for surgical treatment, but, in some small-sized studies, an OS resolution is reported in 15–20% of the cases after two or three cycles [14]. After medical therapy failure, the surgical approach is required to remove the cause of the pathology, to allow the correct drain and ventilation of the sinus, and to prevent disease recurrence. This consists of a functional endoscopic sinus surgery (FESS) and/or oral approach, depending on the type of oral etiology and the condition of the maxillary sinus floor [15].

The aim of this retrospective study is to report the experience of the department of integrated therapies in otolaryngology (Campus Bio-Medico Foundation, Rome, Italy) in classifying and treating patients affected by odontogenic sinusitis derived from “classic complications”, defined as “classical odontogenic complication” in the classification proposed by Felisati et al. in 2013 [9].

## 2. Materials and Methods

Clinical data from the patients surgically treated for odontogenic sinusitis in the department of integrated therapies in otolaryngology (Polyclinic Campus Bio-Medico, Rome, Italy) from December 2020 to June 2021 were considered. A retrospective study was performed, analyzing the collected clinical (etiology, treatment, etc.) and demographic (age, sex, etc.) data.

The inclusion criteria applied revisiting the clinical cases were: (1) clinical diagnosis of sinusitis with suspected odontogenic etiology, supported by radiological and/or endoscopic findings and with medical treatment resistance; (2) specialist (ENT/maxillofacial surgeon or dentist) agreement on the odontogenic focus; (3) surgical treatment via FESS and/or oral approach; (4) presence of computer tomography (CT) executed before the surgery; (5) and SCDDT defined as “classical odontogenic complication”, according to the Felisati classification [9].

Medical treatment resistance was defined as symptoms that were persistence after at least two cycles of oral antibiotics.

Patients presenting history of chronic rhinosinusitis, with or without nasal polyps (CRSwNP or CRSsNP), before the dental condition were excluded.

Of all the patients treated for sinusitis, 68 conformed to the criteria and were included in the study.

To achieve the clinical diagnosis of odontogenic sinusitis with classic complications, we were considering signs and symptoms typically associated with this pathology (purulent rhinorrhea (anterior and/or posterior); for example, unilateral or bilateral nasal obstruction and maxillary pain) that appeared after dental pathology or procedures and did not responding to medical treatment, usually consisting of topical nasal decongestants or steroids, mucolytic therapy, and systemic antibiotics [1,15].

During the first ENT examination, nasal endoscopy was performed using a flexible endoscope with a sterile single-use envelope to explore the nasal cavity and the sinuses; an accurate oral exam was carried out to identify oral or dental lesions; and maxillofacial CT scan (without medium of contrast, MoC) was analyzed to confirm the diagnosis.

The patients were distinguished using the presence or absence of the OAC, respectively, and were classified as type 3A and 3B by Felisati et al. [9]. This distinction guided the surgical treatment unitedly with other considerations, such as the presence of foreign body in the maxillary sinus.

FESS was performed under general anesthesia (GA) using rigid 0°, 30°, 45°, and 70° endoscopes, and the procedure was preceded by nasal decongestion using cottonoids that were soaked with carbocaine and adrenaline, 1: 200.000, to reduce intraoperative bleeding and mucosal congestion. Any significant anatomical variation causing reduced drainage (deviated nasal septum, concha bullosa, etc.) was corrected surgically; uncinectomy and then antrostomy followed, and an eventual foreign body in the maxillary sinus was removed. Other procedures to establish the normal ventilation of the nasal sinuses were performed when necessary. When needed, an oral approach was performed, especially when OAC, a foreign body, or the cause of infection were still present. If an OAC was present, an approach through the fistula was preferred; otherwise, the mini-Caldwell-Luc approach was the treatment of choice.

Patients were discharged after 1 or 2 days, depending on the extension of the surgical treatment. After the surgery, intravenous antibiotics during the hospitalization (1 g cefazolin twice per day) and oral antibiotics at home were recommended (1 g of amoxicillin with clavulanic acid twice per day for 7 days. 

Patients were invited to use nasal washes with 0.9% Na solution at least thrice per day in the first post-operative month. Oral anti-histaminic drugs, such as bilastine or fexofenadine hydrochloride, and nasal oils were encouraged to alleviate the initial post-operative nasal discomfort.

The absence of symptoms and clinical and endoscopic sinusitis signs in the post-operative controls at 10, 30, and 90 days after the intervention was defined as treatment success.

## 3. Results

A total of 68 patients responded to the criteria and to the definition as classical odontogenic complications and were included in this retrospective study; of these, 40 were men (59.7%) and 27 were women (40.3%). The mean age was 45.8 years, (the youngest patient was 33 and the oldest 68). All the patients reported a history of previous dental disease or treatment, and the dental origin of the sinusitis was confirmed by anamnestic record, clinical observation, and radiological evidence. Maxillary pain, nasal obstruction, and an anterior or posterior nasal drip were the most referred symptoms. All patients were resistant to medical therapy. All patients underwent a head CT scan before the intervention. All patients had unilateral sinus opacification. A total of 30 patients (42%) reported an obstruction of the osteo-meatal complex (OMC). A foreign body in the maxillary sinus was found in 9 patients (13%) (Figure 1) and an OAC was present in 27 patients (40%) (Figure 2, Figure 3 and Figure 4). The surgical therapy consisted of a combined oral and nasal simultaneous approach for 28 patients (43%), a combined non-simultaneous approach for 4 patients (6%), a nasal only approach for 14 patients (21%), and an oral only approach for 20 patients (30%) (Figure 5).

Surgical treatment was decided depending on the presentation: an oral approach was preferred when the noxa of the dental infection was still present despite adequate medical therapy; a nasal approach was preferred for patients with an obstruction of the osteo-meatal complex and a presence of predisposing anatomical factors that can hamper the healing of the maxillary sinus; and a combined therapy was preferred when both of these factors were present.

As for standard perioperative and post-operative prophylaxis, all patients received 2 g of intravenous cephazolin during the surgical procedure and the first post-operative day. All patients were discharged on the second post-operative day and received a prescription for 1 g of amoxicillin and clavulanic acid, twice per day for 6 days.

No post-operative TC was performed in order to reduce the radiation exposure of the patients and because the success of the treatment was determined through symptom resolution and clinical inspection. No procedure involved the posterior ethmoid, frontal, or sphenoid sinuses. No procedure had intra-operative or post-operative complications or required revision intervention. All the patients presented a complete resolution of the symptoms at the 30- and 90-day post-operative controls.

## 4. Discussion

A recent increase in OS incidences has been reported; this could be linked to an increase in dental procedures and so, of its complications. This widening sector needs clear and validated protocol, to assure a high standard of care for the patients [16].

In the 2022 European Position Paper (EPOS), importance of a multidisciplinary approach to the assessment and treatment of OS was underlined. It reported that 20% of patients with OS receive a wrong diagnosis, and that 33% get the correct treatment [11]. Antibiotics alone result to be ineffective in a majority of OS cases, requiring surgical treatment [15]. A clear method to predict the efficacy of medical treatment alone is yet to be developed [15].

Multi-microbial infection is the most common type of OS, so the first line of medical treatment consists of ampicillin or piperacillin combined with a β-lactamase inhibitor. Another treatment possibility is a combination of levofloxacin and vancomycin, when the antibiogram is not present [7,8,9,10,11].

A much larger microbiological burden of OS than other forms of sinus inflammation has been demonstrated in the literature [17]. It is also described and reported that there is a higher presence of anaerobic bacteria (e.g., Gram-negative Streptococcus spp., Peptostreptococcus spp., and Fusobacterium spp.) in OS [18]. This is in agreement with the prevalent microbiological findings in dental and peridental infections [19]. Aerobes bacteria such as Streptococcus spp. and Staphylococcus spp. are also reported to be observable in 75% of cases of OS [18].

The presence of a fungal infection (Aspergillus supp.) has been observed in cultures from OS [20].

No microbial culture or antibiogram were performed during the assessment of this pathology because these elements were not considered relevant to the treatment selection, considering that all the patients included in the study were already treated with antibiotic therapy without a resolution of the sinusitis. Thus, we evaluated that surgical therapy is the only way to resolve the pathology. This theory is confirmed by several case series that reported patients failing multiple courses of oral antibiotics before undergoing definitive dental treatment or ESS [21,22,23,24], following the consensus on management. In this way, we avoid long-term antibiotic therapies that can lead to side effects and the development of antibiotic resistance.

The dental treatment is a critical part of the therapy, and the type of dental procedure differs based on the type of dental condition causing the pathology in the first place [25]. The capacity of dental treatment alone to treat OS is debated, and it depends on the nature of the interaction between the maxillary sinus and the dental condition [15]. When both the dental and sinus procedure are performed, it is reported that the order of execution does not influence the success rate [23].

Several types of sino-nasal surgeries have been described, depending on the condition of the maxillary sinus, the osteo-meatal complex, the presence of foreign bodies, and the condition of the other sinuses. One of the most commonly used techniques is the Caldwell–Luc and its modified versions [26]. Endoscopic surgery is adopted when the pathology extends to other sinuses or when there are contraindications to the Caldwell–Luc technique [13].

The clinical elements and treatments for this patient contribute to a better understanding of the Felisati et al. classification and how to correctly use this clinical instrument. The different successful approaches and different clinical conditions of patients belonging to the same group suggest that the Felisati et al. classification is useful but is not sufficient on its own to fully determine the approach to the OS.

The patients, in most of cases, came in for an ENT visit because of their symptoms, and in other cases, they were suggested to undergo an ENT examination by a dentist after a dental scan exam or after the arise of rhinosinusitis symptoms in the post-operative period after a dental implant surgery.

The clinical examination of the patient with suspected OS must include a mouth and nose inspection [27]. Usually the first manifestation of this condition is a nasal obstruction, while dental pain is often absent [27]. Nasal endoscopy is a useful diagnostic tool [28], but in our experience, radiological findings of maxillary sinus inflammation were present independently from mono-lateral purulent secretion coming from the osteo-meatal complex.

To complete the diagnostic workup, it is necessary to perform a radiological investigation. Radiological procedures performed in an odontoiatric office, such as dental scan, can be useful to guide the diagnosis [29], but a full head CT scan is needed to get enough information to proceed with a safe surgical intervention [30]. A Cone Beam CT scan (CBCT) is reported to provide a higher resolution at lower radiological exposition and cost [31].

No major post-operative complication was reported in our group.

The choice of performing a nasal, oral, or combined approach is based on the presence of anatomical elements that facilitate sinusitis and reinfection occurrence, such as adeviated nasal septum, concha bullosa, or an obstructed osteo-meatal complex.

## 5. Conclusions

The data collected and reported in this study confirm that odontogenic sinusitis from classical complication is a condition not solely of the mono-lateral maxillary sinusitis. The involvement of the antrum, the presence of a fungal infection, and the presence of a foreign body are all elements that add complexity to this condition and justify a cautious approach. The correct use of validated classification, the pre-operative CT scan, a multidisciplinary approach, and an appropriate presurgical examination are necessary elements to obtain a good success rate.

## Figures and Tables

**Figure 1 antibiotics-12-00390-f001:**
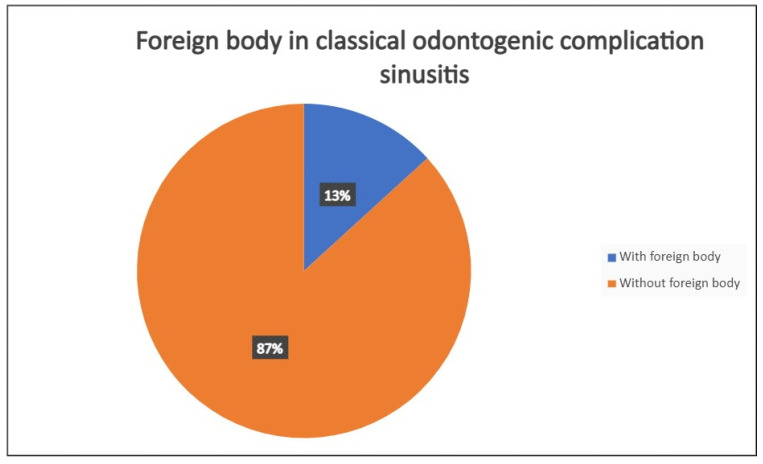
Incidence in presentation of patients with classical odontogenic sinusitis, with or without foreign body in maxillary sinus.

**Figure 2 antibiotics-12-00390-f002:**
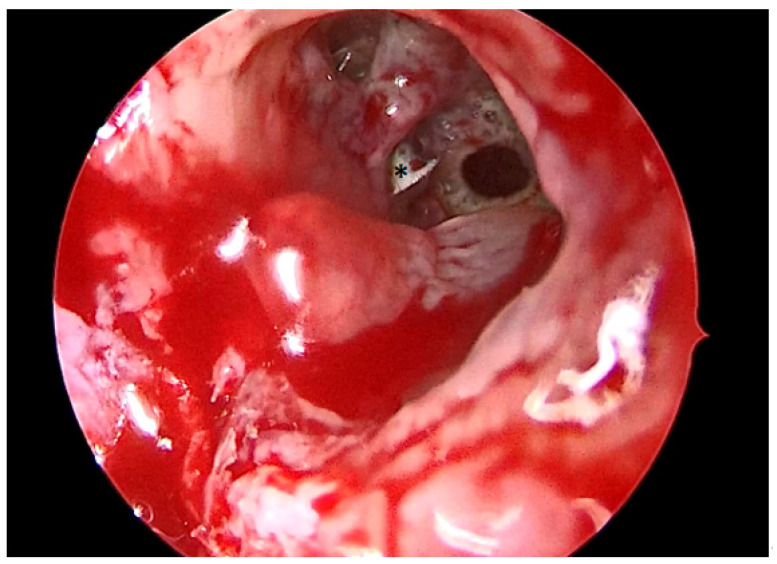
Intraoperative photo of the maxillary sinus seen from the OAF. * = Maxillary foreign body.

**Figure 3 antibiotics-12-00390-f003:**
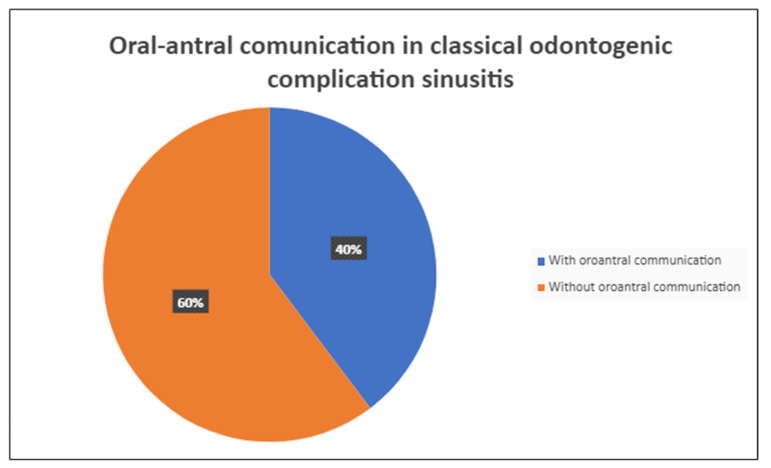
Incidence in presentation of patients with classical odontogenic sinusitis, with or without oro-antral communication in maxillary sinus.

**Figure 4 antibiotics-12-00390-f004:**
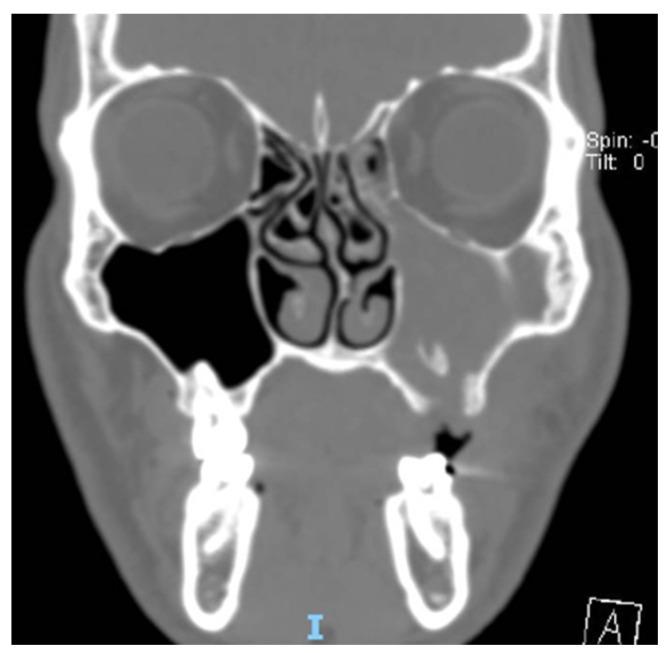
TC exam of patient with OAF and foreign body in maxillary sinus body in maxillary sinus.

**Figure 5 antibiotics-12-00390-f005:**
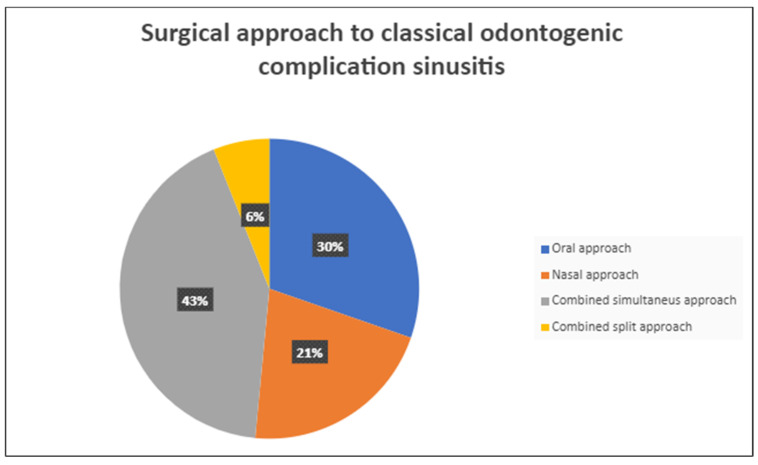
Surgical approach to classical odontogenic sinusitis.

## Data Availability

Not applicable.

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
