# Peer review of "Odontogenic Sinusitis from Classical Complications and Its Treatment: Our Experience"

_antibiotics, 2023, doi:10.3390/antibiotics12020390_

Round 1

Reviewer 1 Report

The results have no statistical relevance and should be improved.

The conclusions are general and not clearly correlated with the results.

Author Response

Dear Reviewer, Thank you for you criticism. The results are a description of the outcome of the treatment of patients affected by odontogenic sinusitis, based on our experience, in a similar way of one of the most cited works in the field, “Felisati G, Chiapasco M, Lozza P, Saibene AM, Pipolo C, Zaniboni M, et al. Sinonasal complications resulting from dental treatment: Outcome-oriented proposal of classification and surgical protocol. American Journal of Rhinology and Allergy. 2013;27”

We corrected the conclusions according to your criticism, explaining better how we arrived to these results.

Reviewer 2 Report

Dear authors, 

I appreciate the topic of the article as being of great interest due to the clinical aspects followed in the study. The material is well structured and very clear, nevertheless, I would suggest introducing some relevant x-ray images in the study, as a support for choosing the treatment modality (which depends, as you described, on soem anatomical elements)and also clinical pictures showing the combined oral and nasal approach . I consider this would increase the quality of the article and would be helpful for young practioners also. 

It is not very clear from the material what is the criteria used in order to associate the oral approach: "When needed oral approach was performed". Please indicate more clearly the indication for every approach and how the treatment modality choice was made.

Author Response

Thank you for your suggestions. We added to the manuscript radiological and clinical images as suggested.

We also clarified in the results what led to the treatment with oral approach.

Reviewer 3 Report

The authors report their experience in the managment of odontogenic sinusitis. Although their results are good, there are no new elements in the present work. Moreover, the paper lacks of significant information (i.e. microbiological study) to be suitable for publication in "antibiotics".

Author Response

Thank you for your suggestions.  We added in the results useful informations on how to choose the correct approach for the treatment of odontogenic sinusitis.

We did not performed additional study on because in our experience this did not change our approach and our results. In fact, the patients included in the study were patients with odontogenic sinusitis resistant to conventional antibiotic therapies, that necessitated surgical therapy, according with the literature. Anyhow, we explained better our approach adding citation to the literature that supports our treatment and justifying better our management.

Reviewer 4 Report

This represents a retrospective descriptive study of sinus complications post dental disease or dental intervention. 

I would suggest that the authors expand their discussion on the following items:

1. Describe the antibiotic protocol or non-surgical intervention that was used prior to deciding the surgical intervention in the study group.

2. Was there any post-surgical treatment imaging to confirm the resolution of sinus disease in this group of 68 patients. I mentioned this with respect to the statement by the authors that all patients had complete resolution of symptoms post treatment but did this also mean radiologic resolution of the disease.

3. It would be interesting to expand on the microbiology of the odontogenic sinusitis and specifically to discuss the presence of fungal sinusitis that was also mentioned by the authors.

4. It might be interesting, although not completely essential, that the authors discuss the prevalence of sinus complications after certain dental interventions.

Author Response

Dear reviewer, thank you for your suggestions. we corrected the manuscript accordingly.

In particular:

1 as suggested we reported the medical therapy that the patients followed before surgical treatment,

2 no radiological imaging was deemed necessary after the intervention because of the symptoms resolution and because it was possible to monitor the patients clinically during the post-op with oral examination and flexible nasal videoendoscopy.

3 we expanded in the discussion the data regarding microbiological population in odontogenic sinusitis

4 thank you for your suggestion, we specified the most common causes of classical complication and provided a suited citation

Round 2

Reviewer 1 Report

The paper is acceptable for publication

Author Response

Thank you for your evaluation

Reviewer 3 Report

The authors partially improved the quality of the paper by adding some concepts to the discussion section. But really the only preoperative medical therapy was 1 gr of amoxicillin and clavulanic acid twice per day in all cases? The authors have never employed quinolones or other antibiotics? 

I'm really convinced that surgery is the mainstay of treatment in many cases of odontogenic sinusitis, but I think that 1 gr of amoxicillin and clavulanic acid every 12 h is sometimes not adequate, indeed the current practice is to prescribe amoxicillin-clavulanate 1 gr every 8 h. In odontogenic sinusitis also quinolones (i.e. moxifloxacin) results effective and worth to be tried before surgery.

In my opinion this paper is still not suitable for publication in "antibiotics"

Author Response

Dear reviewer,

Thank you for you observation. We agree with you, actually the therapy we referred to as preoperative medical therapy is a preparatory therapy, to be used before the surgery to reduce the infection and inflammation to prepare patients to surgery, but it was mistakenly referred in the paper as the only therapy made by the patient before the indication for surgery.

In our clinical practice, we are a secondary care center so usually the patient is referred to us on indication of their dentist, after multiple antibiotic therapies.

We corrected this mistake in the paper, clarifying the antibiotic therapy, and provided citation according to guidelines

Reviewer 4 Report

Structure of sentences still requires some revision. 
There needs to be some additional editing of the English text.  
Reference #7 should be clarified.  

Author Response

Thank you for your suggestion. We corrected reference n°7, edited some English typos and corrected some sentences as suggested.

Round 3

Reviewer 3 Report

The authors fixed the problem highlighted in my previous review; although previous medical therapies are not extensively discussed, it is clear that patients underwent surgery after several cycles of antibiotics

Reviewer 4 Report

Very basic descriptive study of odontogenic sinusitis.  Revised manuscript is certainly improved.